# Legal Framework for Rear-End Crashes in Mixed-Traffic Platooning: A Matrix Game Approach

Xu Chen 🄳 and Xuan Di *🄳

Department of Civil Engineering and Engineering Mechanics, Columbia University, New York, NY 10027, USA
* Correspondence: sharon.di@columbia.edu

**Abstract:** Autonomous vehicles (AV) hold great potential to increase road safety, reduce traffic congestion, and improve mobility systems. However, the deployment of AVs introduces new liability challenges when they are involved in car accidents. A new legal framework should be developed to tackle such a challenge. This paper proposes a legal framework, incorporating liability rules to rear-end crashes in mixed-traffic platoons with AVs and human-propelled vehicles (HV). We leverage a matrix game approach to understand interactions among players whose utility captures crash loss for drivers according to liability rules. We investigate how liability rules may impact the game equilibrium between vehicles and whether human drivers' moral hazards arise if liability is not designed properly. We find that compared to the no-fault liability rule, contributory and comparative rules make road users have incentives to execute a smaller reaction time to improve road safety. There exists moral hazards for human drivers when risk-averse AV players are in the car platoon.

**Keywords:** rear-end crash; liability; mixed traffic





## 1. Introduction

Autonomous vehicles (AV) are anticipated to improve mobility, traffic safety, and accessibility [1]. In the near future, however, AVs will operate on public roads in mixed traffic and will have to manage complex interactions with road users in a traffic environment. Existing work mainly focuses on two polar scenarios where either a single AV navigates in traffic dense with human drivers, or AVs dominate the road [2–5], with negligible interaction with human-controlled counterparts. Much less attention has been afforded to the far more realistic yet challenging transition path between these two scenarios, i.e., when AVs and conventional human-propelled vehicles (HV) must co-exist and interact in a traffic platoon. In addition, those studies on car-following controller designs [6–8] are primarily focused on increasing traffic efficiency. None of them are concerned with the safety aspect of AVs in car-following scenarios. To ensure an equitable transportation ecosystem for both AV and non-AV users, legal liability design in car crashes involving AVs should be discussed.

This paper leverages a matrix game approach to capture the strategic interactions between AVs and HVs in a platoon. In the game environment, one's cost function is a trade-off between traffic safety and efficiency. Upon a good understanding of both AV and HV equilibrium behaviors in the developed game, we would like to further explore human drivers' moral hazards, which are incurred by the presence of AVs. All the existing studies have focused on modeling AVs' new behaviors, but they ignore human drivers' behavioral adaptations to AVs, as humans are exposed to increasing amounts of traffic encounters with AVs. Human drivers may have a weaker incentive to exercise "due care" when faced with AVs. Since human drivers perceive AVs as super-intelligent agents with the ability to adapt to more aggressive and potentially dangerous human driving behavior, the so-called "moral hazard" effect may lower a human driver's caution. It is a well-studied phenomenon in economics [9,10], and is also observed in traffic contexts [11–13].

### 1.1. Related Work

There are many studies leveraging game theoretical approaches to understand car crashes. The level of a driver's precaution (i.e., care level) is adopted to capture the utility function of players in a road safety game [9,10]. To understand the impact of AVs on a transportation system, the road safety game is extended to a game theoretic framework with multiple players in the system: HVs, AVs, AV manufacturers, and law makers [14]. However, these works do not specify the decision making regarding drivers' care levels in any real-world traffic scenarios. An evolutionary game [11] is proposed to study the game equilibrium regarding drivers' reaction times and headway in a car-following model [15,16]. The trajectory data of vehicles are used to quantify the crash severity related to drivers' collision speeds in the utility function. Inspired by the road safety game in car-following models [12], this paper aims to investigate the rear-end crash game in mixed-traffic platoons, where crash loss for drivers is determined by their liabilities [17].

Liability rules utilize insurance principles [18,19] to distribute the internal loss of an accident between parties directly involved in it. In car crashes, drivers' negligence is adopted to measure their liabilities [20,21]. Contributory and comparative negligence are the two main categories. The contributory rule [22] identifies the regime of negligence and non-negligence for drivers, which is used in North Carolina and Virginia. The comparative rule [23] measures the importance of drivers' liabilities according to their contributions to a crash [24]. An empirical analysis [25] shows that drivers have a greater incentive to exercise due care under contributory negligence than under comparative rule. The existing negligence-based liability policies are primarily designed to regulate human drivers' risk-prone behaviors. However, with the emergence of AVs, many legal experts anticipate that driver liability will shift to product liability [26]. In mixed traffic, both drivers' liability and products' liability are needed to regulate AV driving algorithms and a human's driving behavior [27,28]. Liability rules can help AV manufacturers to design operating systems, satisfying their tort obligation [29]. In this work, we will make a comparison of three different liabilities rules in the rear-end crash game.

### 1.2. Contributions of This Paper

The contributions of this paper include:

1. We propose a legal framework that incorporates liability rules to the rear-end crash problems in mixed-traffic platoons.
2. We leverage a matrix game approach for the rear-end crash problem to model interactions in three vehicle-encounter scenarios: HV-HV, AV-HV, and AV-AV scenarios.
3. We perform sensitivity analysis and investigate what factors may impact the equilibrium results of the rear-end crash game.

The rest of this paper is organized as follows: In Section 2, we first present preliminaries regarding the rear-end crash problem and liability rules. In Section 3, we introduce a matrix game to model different scenarios in a rear-end crash. In Section 4, we conduct several numerical experiments. Section 5 concludes and discusses the future work.

## 2. Preliminaries

### 2.1. Rear-End Crash in a Platoon

In this subsection, we briefly introduce a car platoon model in which brake-to-stop events happen [15]. In the platoon (Figure 1), the index of each vehicle is denoted by $0, 1, \ldots, N$. The leading car 0 starts to decelerate with a brake rate $a_0$. After $r_1$ seconds, car 1 starts to decelerate with brake rate $a_1$. $r_1$ denotes the reaction time of car 1 in response to car 0. Similarly, car $i = 2, \ldots, N$ starts to decelerate with brake rate $a_i$ after $r_i$ seconds

in response to the front car $i - 1$. The initial velocity of each vehicle $i$ when starting to decelerate is denoted by $v_i$. Mathematically, the available stopping distance $d_i$ for car $i$ is

$$d_i = \frac{v_0^2}{2a_0} + \sum_{k=1}^{i} v_k(h_k - r_k), i = 1, 2, \ldots, N. \tag{1}$$

$h_i, i = 1, \ldots, N$ denotes the headway (i.e., time distance) between car $i$ and $i - 1$. If the available stopping distance for car $N$ is smaller than its actual stopping distance, a rear-end crash would happen between car $N - 1$ and $N$. We assume there is no crash among car $0, \ldots, N - 1$. We have

$$d_N \leqslant \frac{v_N^2}{2a_N} \tag{2}$$

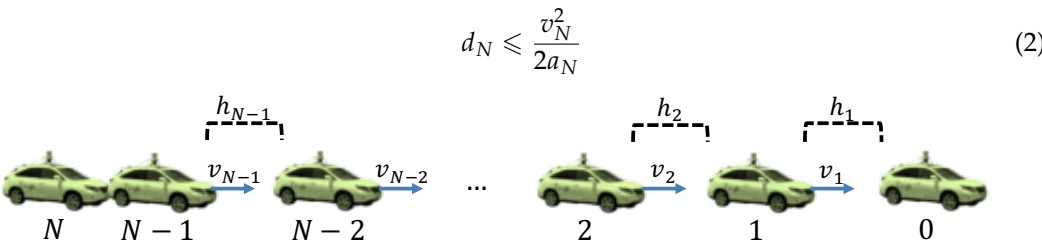

**Figure 1.** Car platoon.

We now look into the crash loss between car $N - 1$ and $N$. Ref. [12] utilizes collision speed $v_c$, i.e., the instantaneous speed of a car when crashing into its front car, to capture the loss function. Mathematically,

$$L = v_c^2 = v_N^2 - 2a_N d_N$$
$$= v_N^2 - 2a_N \Big[\frac{v_0^2}{2a_0} + \sum_{k=1}^{N} v_k(h_k - r_k)\Big]. \tag{3}$$

Equation (3) indicates that the crash loss between car $N - 1$ and $N$ is related to the reaction time and headway of car $1, \ldots, N$. It is shown that as the initial speed $v_i$ goes up, the crash loss increases because a large initial speed usually leads to a large collision speed. In the following section, we will introduce liability rules which utilize drivers' reaction times to determine the crash loss for cars $N - 1$ and $N$.

### 2.2. Liability Rule

We now introduce three liability rules to assign the crash loss between cars $N - 1$ and $N$: no-fault, contributory, and comparative rules. We denote the proportion of crash loss assigned to cars $N - 1$ and $N$ as $S_{N-1}$ and $S_N$, respectively. We have $S_{N-1} + S_N = 1$.

1.  No-fault rule [30]: The rule was first utilized to determine crash loss between automobiles in New York state. It is applied to any cyclist, pedestrian, passenger, or driver injured by a motor vehicle. There are now 12 states adopting the no-fault rule. The crash loss $L$ is assigned to drivers, regardless of who is at fault in a car crash. Mathematically,

$$S_{N-1} = S_N = \frac{1}{2} \tag{4}$$

2.  Contributory rule [22]: The crash loss $L$ is assigned to drivers according to the regime of negligence and non-negligence. We adopt the reaction time $r$ to define the regime. Mathematically,

$$\begin{cases} S_{N-1} = 0, S_N = 1, if \ r_{N-1} < \bar{r}, r_N > \bar{r} \\ S_{N-1} = 1, S_N = 0, if \ r_{N-1} > \bar{r}, r_N < \bar{r} \\ S_{N-1} = S_N = \frac{1}{2}, if \ (r_{N-1} - \bar{r})(r_N - \bar{r}) \geqslant 0 \end{cases} \tag{5}$$

$\bar{r}$ is a baseline to identify negligence and non-negligence conditions according to drivers' reaction times. In this work, we assume $\bar{r} = 1.5$ s, which is the average reaction time in brake-to-stop events obtained from real-world scenarios [12].

3. Comparative rule [23,24]: The crash loss $L$ is assigned to drivers according to their contributions to the rear-end crash. In this work, we use reaction time to measure drivers' contributions to a car accident. Mathematically,

$$S_{N-1} = \frac{e^{r_{N-1}}}{e^{r_{N-1}} + e^{r_N}}, S_N = \frac{e^{r_N}}{e^{r_{N-1}} + e^{r_N}} \tag{6}$$

## 3. Matrix Game Approach

In this section, we introduce a matrix game approach to model the rear-end crash in a car platoon. Assumptions we will use for the game approach are first presented:

### 3.1. Assumptions

1. The rear-end crash only happens between two vehicles. Crashes among three or more vehicles are not considered in this work.
2. Vehicle $1, \ldots, N-2$ are not involved in the crash. They are non-strategic players whose reaction times are predetermined.
3. All vehicles in the car platoon share the same initial velocity, break rate, and headway. We have: $v_i = v_0, a_i = a_0, h_i = \bar{h}, i = 1, \ldots, N$.
4. Players in different encounter scenarios know whether their opponents are HVs or AVs. In other words, an AV's reaction time in one scenario does not affect its choice in other scenarios. This is different from the assumption [14] that the decision making of AVs is predetermined by an AV manufacturer.

### 3.2. Game Formulation

We consider three vehicle encounter scenarios [14] for rear-end crash between car $N-1$ and $N$: an HV encounters the other HV (HH scenario), an AV encounters an HV (AH scenario), and an AV encounters the other AV (AA scenario).

**HH scenario**: The rear-end crash happens between two human drivers. We specify elements in the rear-end crash game as follows:

- *Players*: Human drivers play a symmetric matrix game.
- *Decision variables*: Reaction time measures the level of precaution (i.e., care level [12,14]) for drivers when navigating roads. $r_{N-1}^H \in R^H$ and $r_N^H \in R^H$ represent the reaction time of human drivers for car $N-1$ and car $N$, respectively. $R^H = \{\bar{r}_a^H, \bar{r}_p^H\}$ is a discrete feasible set for players. $\bar{r}_a^H$ indicates risk-averse behavior with a short reaction time, and $\bar{r}_p^H$ is risk-prone behavior with a long reaction time.
- *Utility*: The utility of drivers in the rear-end crash captures the effects of reaction time, and the crash loss assigned to drivers. $U_{N-1}^{(HH)}$ and $U_N^{(HH)}$ represent the utility of drivers in cars $N-1$ and $N$, respectively. We have

$$U_{N-1}^{(HH)}(r_{N-1}^H, r_N^H) = \beta_h \cdot f_h(r_{N-1}^H) + (1 - \beta_h) \cdot (-L \cdot S_{N-1}),$$
$$U_N^{(HH)}(r_{N-1}^H, r_N^H) = \beta_h \cdot f_h(r_N^H) + (1 - \beta_h) \cdot (-L \cdot S_N), \tag{7}$$

where $\beta_h$ and $1 - \beta_h$ are trade-off coefficients in the utility function. $f_h(r)$ is the utility function with respect to players' reaction times. $f_h(r)$ satisfies following properties [14]: (1) $\frac{df_h(r)}{dr} > 0$. (2) $\frac{d^2 f_h(r)}{dr^2} < 0$, indicating that the marginal utility decreases as the reaction time $r$ increases.

- *Payoff Matrix*: Given players' decision variables and utility functions, we can formulate the payoff matrix for cars $N-1$ and $N$ in the HH scenario as follows:

Car $N$

$$\begin{array}{c|c|c}
 & \bar{r}_a^H & \bar{r}_p^H \\
\hline
\bar{r}_a^H & U_{N-1}^{(HH)}(\bar{r}_a^H, \bar{r}_a^H), U_N^{(HH)}(\bar{r}_a^H, \bar{r}_a^H) & U_{N-1}^{(HH)}(\bar{r}_a^H, \bar{r}_p^H), U_N^{(HH)}(\bar{r}_a^H, \bar{r}_p^H) \\
\hline
\bar{r}_p^H & U_{N-1}^{(HH)}(\bar{r}_p^H, \bar{r}_a^H), U_N^{(HH)}(\bar{r}_p^H, \bar{r}_a^H) & U_{N-1}^{(HH)}(\bar{r}_p^H, \bar{r}_p^H), U_N^{(HH)}(\bar{r}_p^H, \bar{r}_p^H)
\end{array}$$

Car $N-1$

- *Game Equilibrium*: At equilibrium, no human drivers can improve the utility by unilaterally changing the reaction time.

  **AH scenario**: We now specify elements in the AH scenario:

- *Players*: An AV and an HV play an asymmetric matrix game. Note that there are two cases in the AH scenario: (car $N-1$, car $N$) is (HV, AV) and (car $N-1$, car $N$) is (AV, HV). For simplicity, we present the case when (car $N-1$, car $N$) is (HV, AV) in this section. In numerical experiments, we investigate both cases.
- *Decision variables*: $r_{N-1}^H \in R^H$ and $r_N^A \in R^A$ denote the reaction time. $R^A = \{\bar{r}_a^A, \bar{r}_p^A\}$ and $\bar{r}_a^A, \bar{r}_p^A$ indicate risk-averse and risk-prone behaviors, respectively.
- *Utility*: The utility of cars $N-1$ and $N$ is given by:

$$U_{N-1}^{(AH)}(r_{N-1}^H, r_N^A) = \beta_h \cdot f_h(r_{N-1}^H) + (1 - \beta_h) \cdot (-L \cdot S_{N-1}),$$
$$U_N^{(AH)}(r_{N-1}^H, r_N^A) = \beta_a \cdot f_a(r_N^A) + (1 - \beta_a) \cdot (-L \cdot S_N), \tag{8}$$

  where $\beta_a$ and $1 - \beta_a$ are trade-off coefficients in the utility function.
- *Payoff Matrix*: Given players' decision variables and utility functions, we can formulate the payoff matrix for cars $N-1$ and $N$ in the AH scenario as follows:

Car $N$

$$\begin{array}{c|c|c}
 & \bar{r}_a^A & \bar{r}_p^A \\
\hline
\bar{r}_a^H & U_{N-1}^{(AH)}(\bar{r}_a^H, \bar{r}_a^A), U_N^{(AH)}(\bar{r}_a^H, \bar{r}_a^A) & U_{N-1}^{(AH)}(\bar{r}_a^H, \bar{r}_p^A), U_N^{(AH)}(\bar{r}_a^H, \bar{r}_p^A) \\
\hline
\bar{r}_p^H & U_{N-1}^{(AH)}(\bar{r}_p^H, \bar{r}_a^A), U_N^{(AH)}(\bar{r}_p^H, \bar{r}_a^A) & U_{N-1}^{(AH)}(\bar{r}_p^H, \bar{r}_p^A), U_N^{(AH)}(\bar{r}_p^H, \bar{r}_p^A)
\end{array}$$

Car $N-1$

- *Game Equilibrium*: At equilibrium, no HV or AV can improve the utility by unilaterally changing reaction time.

  **AA scenario**: We now specify elements in the AA scenario:

- *Players*: Two AVs play a symmetric game.
- *Decision variables*: $r_{N-1}^A, r_N^A \in R^A$ denote the reaction time of AVs.
- *Utility*: The utility of cars $N-1$ and $N$ is given by:

$$U_{N-1}^{(AA)}(r_{N-1}^A, r_N^A) = \beta_a \cdot f_a(r_{N-1}^A) + (1 - \beta_a) \cdot (-L \cdot S_{N-1}),$$
$$U_N^{(AA)}(r_{N-1}^A, r_N^A) = \beta_a \cdot f_a(r_N^A) + (1 - \beta_a) \cdot (-L \cdot S_N). \tag{9}$$

- *Payoff Matrix*: Given players' decision variables and utility functions, we can formulate the payoff matrix for cars $N-1$ and $N$ in the AA scenario as follows:

Car $N$

$$\begin{array}{c|c|c}
 & \bar{r}_a^A & \bar{r}_p^A \\
\hline
\bar{r}_a^A & U_{N-1}^{(AA)}(\bar{r}_a^A, \bar{r}_a^A), U_N^{(AA)}(\bar{r}_a^A, \bar{r}_a^A) & U_{N-1}^{(AA)}(\bar{r}_a^A, \bar{r}_p^A), U_N^{(AA)}(\bar{r}_a^A, \bar{r}_p^A) \\
\hline
\bar{r}_p^A & U_{N-1}^{AA}(\bar{r}_p^A, \bar{r}_a^A), U_N^{(AA)}(\bar{r}_p^A, \bar{r}_a^A) & U_{N-1}^{(AA)}(\bar{r}_p^A, \bar{r}_p^A), U_N^{(AA)}(\bar{r}_p^A, \bar{r}_p^A)
\end{array}$$

Car $N-1$

- *Game Equilibrium*: At equilibrium, no AVs can improve the utility by unilaterally changing reaction time.

**Remark 1.** *1. Note that cars $0, 1, \ldots, N-2$ in the platoon are not involved in the rear-end crash. The available stopping distance for car $i$, $i = 1, \ldots, N-1$ is larger than the actual stopping distance. We have*

$$d_i = \frac{v_0^2}{2a_0} - \sum_{k=1}^{i} v_k(r_k - h_k) \geqslant \frac{v_i^2}{2a_i}, i = 1, \ldots, N-1 \tag{10}$$

*In numerical experiments, we will investigate how the reaction time of cars $1, \ldots, N-2$ may impact the equilibrium results of the rear-end crash game.*

2. *Mixed Nash equilibrium may exist in the rear-end crash game. For example, the mixed Nash equilibrium for an AV is to choose action $\bar{r}_a^A$ with probability $p$ and $\bar{r}_p^A$ with probability $1 - p$. We then use the average policy to denote the equilibrium $r^{A*}$ for the AV, i.e., $r^{A*} = p \cdot \bar{r}_a^A + (1-p) \cdot \bar{r}_p^A$.*

*3.3. Performance Measure*

We now define the performance measure to evaluate the equilibrium results.

**Definition 1.** *(Moral Hazard.) We say that a moral hazard [10] happens to a road user i if the following condition holds:*

$$r^*(x) > r^*(x') \tag{11}$$

*where $r^*$ is the reaction time at the equilibrium, $x$ represents road environment or other road users' behaviors, and $x'$ represents an improved road environment or other road users' behaviors. In other words, a moral hazard happens if a road user chooses a longer reaction time when others' care levels or the road environment are improved.*

**4. Numerical Experiments**

In this section, we conduct numerical experiments and sensitivity analysis in order to understand following research questions:

1. How do liability rules impact the equilibrium results in rear-end crash games?
2. Under what circumstances does a moral hazard exist for human drivers in the platoon?
3. What factors may influence the reaction time of HVs and AVs at equilibrium?

We first look into how the reaction time of non-strategic players (i.e., cars $1, \ldots, N-2$) in the car platoon may affect the equilibrium between car $N-1$ and $N$ in the rear-end crash game. We briefly introduce the set-up of our numerical experiments: $N = 12$, $v_0 = 60$ feet/s, $a_0 = 6$ feet/s$^2$. We assume $\bar{h} = 1.4$ s, which is the average headway in brake-to-stop events obtained from real-word scenarios [12]. Non-strategic players such as car $i$ ($i = 1, \ldots, 10$) can adopt a short reaction time $r = 0.5$ s (i.e., risk-averse behavior) or a long reaction time $r = \min\{\bar{h} + \sum_{k=1}^{i-1}(\bar{h} - r_k), 2.5\}$ s (i.e., risk-prone behavior) according to Equation (10). We consider three vehicle encounter scenarios: HH, AA, and AH scenarios for strategic players, namely car $N-1$ and $N$, which are involved in the rear-end crash. Other parameters are specified as: $\bar{r}_a^A = \bar{r}_a^H = 0.5$ s, $\bar{r}_p^A = \bar{r}_p^H = 2.5$ s. $\beta_h = 0.2$ and $\beta_a = 0.3$, representing the trade-off coefficients of utility functions with respect to reaction time for HVs and AVs, respectively. This indicates: (1) humans require a higher cost than AVs to achieve the same reaction time; (2) AVs have much more perception and reaction power than humans [12,14].

Figure 2 demonstrates the equilibrium results of three vehicle encounter scenarios. We first study the HH scenario. Note that the HH scenario is a symmetric game. We only visualize the reaction time of one HV $r^*$ at equilibrium in Figure 2a for simplicity. The $x$-axis denotes the proportion of non-strategic players (car $1, \ldots, 10$) with reaction time 0.5 s in the platoon. The $y$-axis denotes the reaction time at equilibrium $r^* = p \cdot \bar{r}_a^H + (1-p) \cdot \bar{r}_p^H$ where $p$ is the probability of choosing a short reaction time $\bar{r}_a^H = 0.5$ s in the mixed Nash equilibrium for the HV player. We visualize the probability $p$ in Figure 2b. The blue, red, and green lines denote the game equilibrium with no-fault, contributory, and comparative

rules, respectively. The trend of all lines in Figure 2a shows that as the proportion of vehicles with risk-averse behaviors increases, the reaction time $r^*$ of HV players increases. In other words, the probability of choosing a short reaction time 0.5 s decreases (Figure 2b). We can also observe the trend in other road safety games [10,14]. The interpretation is: when more drivers in the platoon become attentive, strategic player cars $N-1$ and $N$ begin to increase their reaction time and become less cautious by taking advantage of risk-averse vehicles in the car platoon. We then look into the equilibrium results in the AH scenario. In Figure 2e, the reaction time of the AV player is smaller than that of the HV player, indicating that the presence of AV players makes human drivers less cautious. HV players can take advantage of AV players. Given a fixed proportion of risk-averse and risk-prone drivers in the platoon, the available stopping distance for human drivers will increase if there are more AVs choosing a short reaction time. In this case, a moral hazard exists when HV players become less attentive. AVs do not necessarily improve road safety, especially when there exists moral hazards for strategic HV players. The available stopping distance increases as the ratio of risk-averse to risk-prone drivers increases, which makes HV players less attentive with a shorter reaction time. In this case, a moral hazard exists when HV players take advantage of the increasing number of risk-averse drivers.

We make a comparison of three liability rules by investigating the reaction times of players at equilibrium. The red line is aligned with the green line, which means there is no significant difference between contributory and comparative rules. When the proportion of risk-averse vehicles in the platoon is larger than 0.3, the equilibrium results with three liability rules are the same. When the proportion of risk-averse vehicles in the platoon is smaller than 0.3, the reaction time at equilibrium with the no-fault rule is larger than those with the two other rules. The explanation is that the no-fault rule does not quantify drivers' contributions to the rear-end crash, making drivers lack motivation to enhance their level of precaution.

We now look into how headway in the car platoon may impact the equilibrium results of the rear-end crash game. The set-up of our numerical experiments is demonstrated as follows: $N = 12, v_0 = 60$ feet/s, $a_0 = 6$ feet/s$^2$. We vary the value of headway $\bar{h}$ from 1 to 4. We fix the proportion of non-strategic players (car $i$ ($i = 1, \ldots, 10$)) who adopt a short reaction time $r = 0.5$ s (i.e., risk-averse behavior) as 50%. Other parameters remain the same. We still consider three vehicle encounter scenarios: HH, AA, and AH scenarios for strategic player cars $N-1$ and $N$, which are involved in the rear-end crash game.

Figure 3 demonstrates the equilibrium results of three vehicle encounter scenarios in the rear-end crash game. The x-axis denotes the headway $\bar{h}$. The y-axis in Figure 3a denotes the reaction time $r^*$ in the mixed Nash equilibrium. The y-axis in Figure 3b denotes the probability $p$ of choosing a short reaction time $\bar{r}_a^H = 0.5$ s for the HV player. When $1$ s $\leqslant \bar{h} \leqslant 1.3$ s, the HV player chooses the short reaction time 0.5 s with probability $p = 1$, meaning that human drivers are attentive. When $1.3$ s $\leqslant \bar{h} \leqslant 2.3$ s, the reaction time at equilibrium has a positive relationship with headway. As the time distance between vehicles increases, drivers become less attentive and increase their reaction time. When $\bar{h} \geqslant 2.3$ s, the HV player chooses the long reaction time 2.5 s with probability $p = 1$. This is because the time distance between vehicles ensures that drivers will not be involved in crashes when $r = 2.5$ s. The trend of all lines in Figure 3a shows that the reaction time of drivers has a strong correlation with the headway in the car platoon. The increase in headway allows drivers to execute a longer reaction time. We can also observe the trend in the AA scenario (Figure 3c) and evolutionary games [12]. The interpretation is that a longer headway in car platoons leads to a safer driving environment where a moral hazard may exist for road users. We then look into the equilibrium results in the AH scenario. In Figure 3e, the reaction time of the AV player is smaller than that of the HV player, indicating that the presence of AV players makes human drivers less cautious. Note that in our paper, the headway in car platoons is a constant. In real-world scenarios, headway is also a decision variable for drivers. We will investigate more complex decision making in platooning in the future.

There are many other factors that may impact the equilibrium results in rear-end crash games: velocity, brake rate, trade-off coefficients in the utility function, and platoon size. We visualize the game equilibrium when varying velocity $v_0$ from 30 to 100 feet/s (Figure 4). In Figure 4a, the blue and red lines represent the reaction time of AVs and HVs in the AH scenario, respectively. When $v_0 \leqslant 40$ feet/s, both AVs and HVs choose a long reaction time $r^* = 2.5$ s. When $40 \leqslant v_0 \leqslant 80$ feet/s, the reaction time at equilibrium decreases as the velocity increases. When $v_0 \geqslant 80$ feet/s, drivers choose a long reaction time. This is because a rear-end crash cannot be avoided when vehicles have a high speed. Similar to headway and velocity, brake rate and platoon size determine the available stopping distance for HVs and AVs in rear-end crashes. A longer headway, along with a smaller velocity and a higher brake rate, leads to a safer driving environment where moral hazards may exist. We visualize the game equilibrium when varying the trade-off coefficient $\beta_h$ (Figure 5). It is shown that the reaction time at equilibrium and $\beta_h$ have a positive relationship. The trade-off coefficients measure the impact of crash loss on HVs' and AVs' utility, which is related to the external road environment and government subsidies [14].

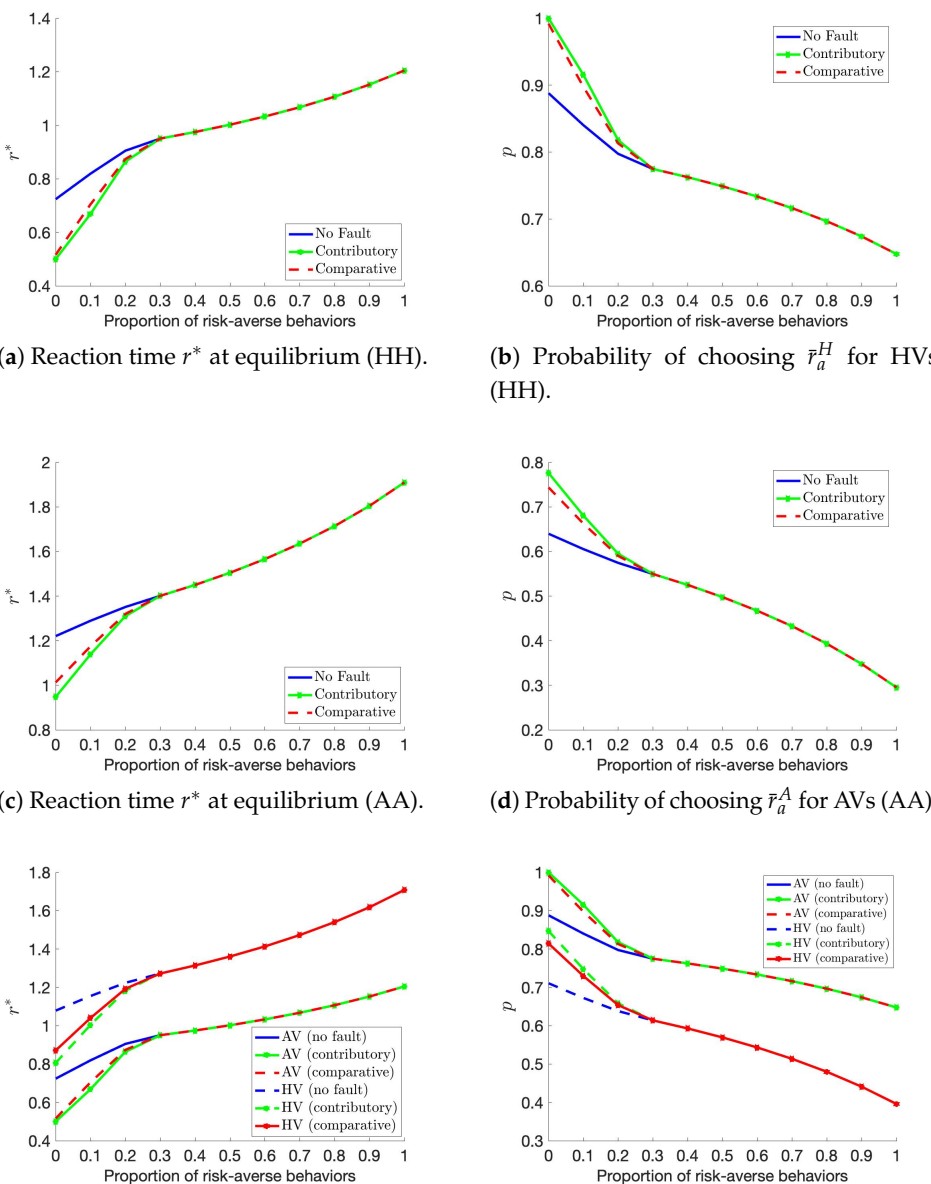

(**a**) Reaction time $r^*$ at equilibrium (HH).

(**b**) Probability of choosing $\bar{r}_a^H$ for HVs (HH).

(**c**) Reaction time $r^*$ at equilibrium (AA).

(**d**) Probability of choosing $\bar{r}_a^A$ for AVs (AA).

(**e**) Reaction time $r^*$ at equilibrium (AH).

(**f**) Probability of choosing $\bar{r}_a^A$ and $\bar{r}_a^H$ (AH).

**Figure 2.** Non-strategic players.

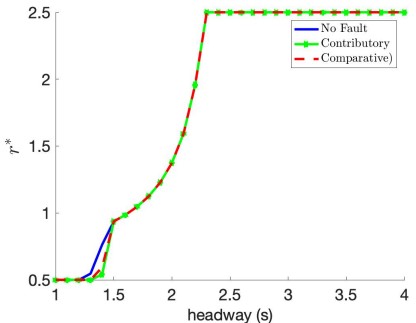
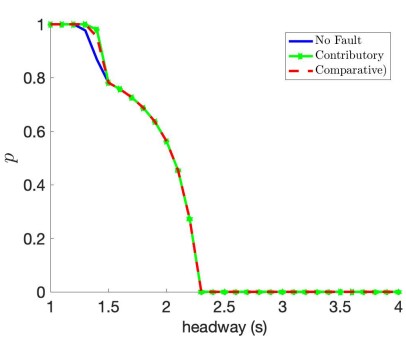

(**a**) Reaction time $r^*$ at equilibrium (HH).

(**b**) Probability of choosing $\bar{r}_a^H$ for HVs (HH).

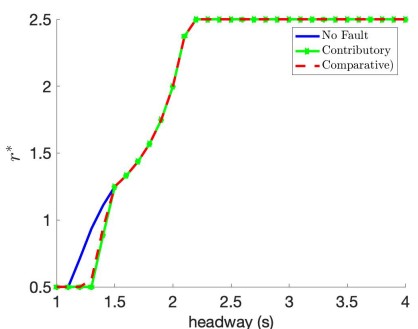
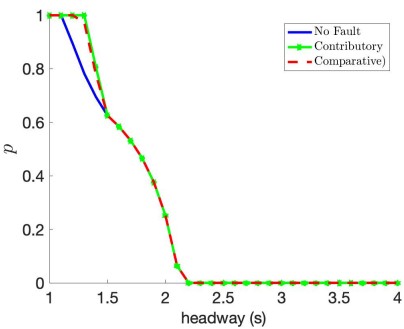

(**c**) Reaction time $r^*$ at equilibrium (AA).

(**d**) Probability of choosing $\bar{r}_a^A$ for AVs (AA).

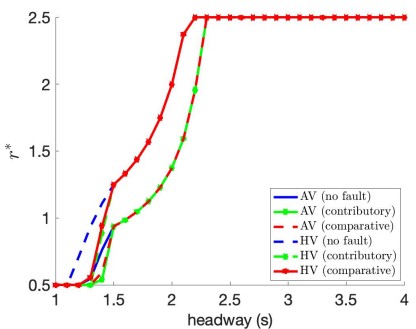
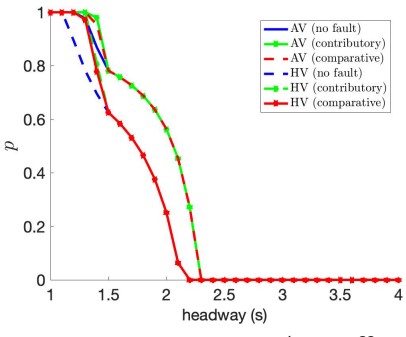

(**e**) Reaction time $r^*$ at equilibrium (AH).

(**f**) Probability of choosing $\bar{r}_a^A$ and $\bar{r}_a^H$ (AH).

**Figure 3.** Headway.

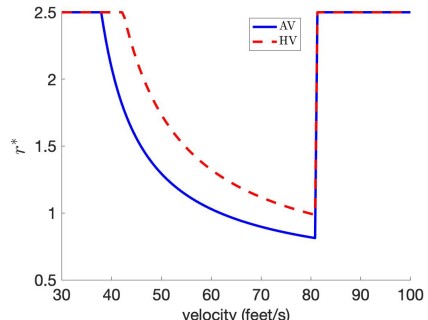

(**a**) Reaction time $r^*$ at equilibrium.

(**b**) Probability of choosing $\bar{r}_a^A$ and $\bar{r}_a^H$.

**Figure 4.** Velocity.

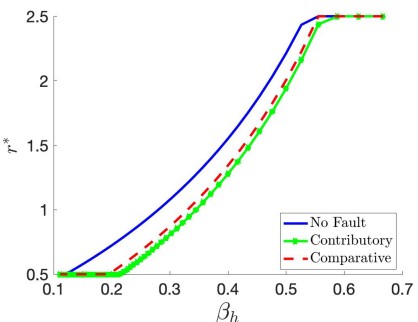
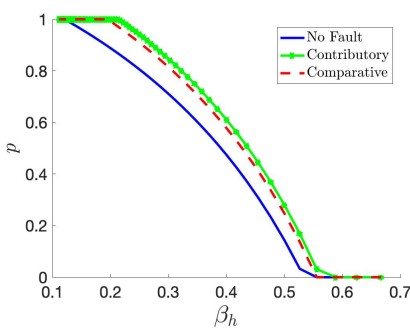

(**a**) Reaction time $r^*$ at equilibrium.  (**b**) Probability of choosing $\bar{r}_a^H$ .

**Figure 5.** Trade-off coefficient $\beta_h$.

We summarize the numerical results as follows:

1. There exists moral hazards for human drivers if risk-averse drivers are in the platoon. This is mainly because risk-averse drivers enlarge the available stopping distance, making drivers in the following vehicles less cautious in brake-to-stop events.
2. Compared to HVs, AVs execute a smaller reaction time in rear-end crashes, indicating that AVs are more conservative than HVs. Human drivers tend to be less attentive by increasing their reaction time when encountering AVs.
3. Compared to the no-fault rule, contributory and comparative rules make road users have more incentives to reduce their reaction time and improve the road safety in platooning.
4. The reaction time at equilibrium has a positive relationship with the headway in the car platoon. A longer headway creates a safer driving environment where a longer reaction time can be executed.

## 5. Conclusions and Discussion

This paper proposes a legal framework for rear-end crashes to understand drivers' liabilities in mixed-traffic platooning. We utilize a matrix game approach for various encounter scenarios among HVs and AVs in rear-end crashes. The utility function in the matrix game captures the effects of reaction time and the crash loss assigned to drivers based on three liability rules. We conduct numerical examples to investigate the equilibrium results of rear-end crash games.

Our findings are summarized as follows: (1) Risk-averse AVs increase the available stopping distance in brake-to-stop events, creating a safer driving environment where human drivers can execute a longer reaction time. Moral hazards exist when the proportion of risk-averse AVs in the car platoon increases. (2) The reaction time of drivers at game equilibrium has a strong correlation with the headway between vehicles. Drivers become less attentive when the headway increases. (3) Contributory and comparative rules perform better than the no-fault rule on improving road safety because drivers have more incentive to execute a shorter reaction time when their contributions to rear-end crashes are considered into liability rules.

We briefly discuss the limitations of this work: (1) The decision making of drivers in the platoon is simplified as the reaction time. There are many other decision variables for players, including velocity, brake rate, and headway. (2) The individual utility function in this work does not consider external road environments and social effects. It is challenging to quantify external road environments in the utility function.

This work can be extended in following ways: (1) Liability rules for rear-end crashes should be modified to minimize social cost, including road safety and travel efficiency. We will develop a solution approach to identify the best regime of negligence and non-negligence conditions for drivers. (2) A more complicated game (i.e., *N*-player differential asymmetric game), in which all vehicles in the platoon make optimal decisions, should

be proposed to understand interactions among all HVs and AVs. (3) Crash data from real-world scenarios should be utilized to calibrate parameters in the rear-end crash model, including crash loss and crash probability. This work can also be extended to other car accidents in real-world scenarios, such as in angle crashes and sideswipe crashes.

**Author Contributions:** Conceptualization, X.C. and X.D.; methodology, X.C.; validation, X.C.; writing—original draft preparation, X.C.; writing—review and editing, X.D.; visualization, X.C.; supervision, X.D.; project administration, X.D.; funding acquisition, X.D. All authors have read and agreed to the published version of the manuscript.

**Funding:** This work is sponsored by NSF under CAREER award number CMMI-1943998.

**Institutional Review Board Statement:** Not applicable.

**Informed Consent Statement:** Not applicable.

**Data Availability Statement:** Not applicable.

**Conflicts of Interest:** We confirm that neither the manuscript nor any parts of its content are currently under consideration or published in another journal. All authors have approved the manuscript and agree with its submission to the journal "Future transportation".

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
