# Peer review of "Legal Framework for Rear-End Crashes in Mixed-Traffic Platooning: A Matrix Game Approach"

_futuretransp, doi:10.3390/futuretransp3020025_

Round 1
Reviewer 1 Report
This paper covers a very interesting and important topic in mixed traffic platooning - who should be responsible for rear-end crashes in platooning? Overall, this study is well-organized, and the results are convincing enough to me. I have a few minor comments that the authors might need to take care of:
1) for sentences, e.g., "[29] demonstrates that liability ...", it is suggested that author names be used at the beginning of sentences of this type and move reference to the end of this sentence.
2) "...driver injured by a motor vehicle in New York state." is this study specific to a region i.e., New York state, or can be applied in other regions too?
3) I see authors have discussed many other factors that may impact the equilibrium results in rear-end crashes. It would great for the authors to discuss more what those impacts look like. Particularly, the authors have made a lot of assumptions, I would also suggest the authors perform some sensitivity analysis to see if anything big will occur in the results.
Author Response
1. for sentences, e.g., "[29] demonstrates that liability ...", it is suggested that author names be used at the beginning of sentences of this type and move reference to the end of this sentence. Response: Thanks for the suggestion. We modified the sentence in the new manuscript.
2. "...driver injured by a motor vehicle in New York state." is this study specific to a region i.e., New York state, or can be applied in other regions too?
Response: Thanks for pointing it out. Yes, it is applied in other regions. We would like to clarify that no fault rule was first utilized to determine crash loss between automobiles in New York state. There are now 12 states adopting this rule (The impact of No-Fault legislation on Automobile Insurance, 2009). We modified our statement in the new manuscript.
3. I see authors have discussed many other factors that may impact the equilibrium results in rear-end crashes. It would great for the authors to discuss more what those impacts look like. Particularly, the authors have made a lot of assumptions, I would also suggest the authors perform some sensitivity analysis to see if anything big will occur in the results.
Response: Thanks for the suggestion. We added sensitivity analysis on the relationship between game equilibrium and velocity, trade-off coefficients in numerical experiments. Please refer to Section 4 in the new manuscript.
Reviewer 2 Report
Very glad to review this paper (futuretransp-2255558). Thanks for your waiting. This paper proposes a legal framework, incorporating liability rules to rear-end crashes in mixed traffic platoons with AVs and HVs, which utilize a matrix game approach for various encounter scenarios among HVs and AVs in rear-end crashes. According to the three liability rules, the utility function in the matrix game can effectively capture the effects of reaction time and the crash loss assigned to drivers. This paper is innovative. The authors fully explain the different performances of different car drivers in rear-end crashes. The overall quality of this paper is great.
Main problems:
i. I notice that the line numbers of the full paper may be wrong.
In Preliminaries:
ii. In Figure 1, it is best to mark the direction of the road.
iii. In L80-L87, I believe that assumptions should be separately described in a subsection "Assumptions", item by item.
In Matrix game approach:
iv. Similarly, in L143-L146, the content described by authors should also be considered as an assumption, and this description can also be included in the subsection "Assumptions".
Author Response
1. I notice that the line numbers of the full paper may be wrong.
Response: Line numbers are predetermined by the latex template of Future Transportation journal when the article is in submission. They will be removed in the final version.
2. In Figure 1, it is best to mark the direction of the road.
Response: Thanks for the suggestion. We added the direction of the road to Figure 1 in the new manuscript.
3. In L80-L87, I believe that assumptions should be separately described in a subsection "Assumptions", item by item.
4. Similarly, in L143-L146, the content described by authors should also be considered as an assumption, and this description can also be included in the subsection "Assumptions".
Response: Thanks for the suggestion. We added a subsection “Assumptions” in Section 3.1 in the new manuscript.